# Impact of Respiratory Syncytial Virus (RSV) in Adults 60 Years and Older in Spain

**DOI:** 10.3390/geriatrics9060145

**Published:** 2024-11-06

**Authors:** Sara Jimeno Ruiz, Adrián Peláez, Ángeles Calle Gómez, Mercedes Villarreal García-Lomas, Silvina Natalini Martínez

**Affiliations:** 1Departamento de Pediatría, Hospital HM Puerta del Sur, HM Hospitales, 28938 Madrid, Spain; sarajimeno@hotmail.com (S.J.R.); acalle@mail.hmhospitales.com (Á.C.G.); slnatalini@hmhospitales.com (S.N.M.); 2Unidad de Vacunas, HM Hospitales, 28938 Madrid, Spain; 3Facultad de Ciencias de la Salud-HM Hospitales, Universidad Camilo José Cela, 28692 Madrid, Spain; mvillarreal@hmhospitales.com; 4Departamento de Unidad de Análisis de Datos, Fundación de Investigación HM Hospitales, 28938 Madrid, Spain; 5Centro de Investigación en Red de Enfermedades Respiratorias (CIBERES), Instituto de Salud Carlos III (ISCIII), 28029 Madrid, Spain; 6Departamento de Medicina Interna, Hospital HM Torrelodones, Hospital HM, 28250 Madrid, Spain

**Keywords:** respiratory syncytial virus, respiratory viruses, elder population, hospitalization, ICU admission, mortality

## Abstract

**Background/Objectives**: Respiratory illnesses frequently lead to hospitalization in adults aged 60 and older, especially due to respiratory viral infectious (RVI). This study investigates hospitalization patterns and characteristics of RVI at HM Hospitals from October 2023 to March 2024; **Methods**: We retrospectively explored hospitalizations of patients aged 60 years and older with RVIs, gathering data on demographics, clinical profiles, comorbidities, and treatments. Outcomes included hospitalization, ICU admissions, and mortality, and independent factors associated with outcomes were identified using a multi-state model; **Results**: From October 2023 to March 2024, from a total of 3258 hospitalizations, 1933 (59.3%) were identified as positive for RVIs. Overall, SARS-CoV-2 was the most prevalent (52.6%), followed by influenza (32.7%), and RSV (11.8%). Most RVI involved single infections (88.2%). Hospitalization rates increased with age for SARS-CoV-2 (333.4 [95% CI: 295.0–375.2] to 651.6 [95% CI: 532.1–788.4]), influenza (169.8 [95% CI: 142.6–200.7] to 518.6 [95% CI: 412.1–643.1]), and RSV (69.2 [95% CI: 52.2–90.0] to 246.0 [95% CI: 173.8–337.5]), with SARS-CoV-2 showing the highest rate, followed by influenza and RSV. In the multi-state model, RSV infection significantly increased ICU admission risk (HR: 2.1, 95%, *p* = 0.037). Age on admission (HR: 1.1, 95%, *p* < 0.001) and Charlson score (HR: 1.4, 95%, *p* = 0.001) were associated with transitioning from admission to death. ICU to death risks included age at admission (HR: 1.7, 95%, *p* < 0.001); **Conclusions**: RVI in adults 60 years and older are associated with high hospitalization and mortality rates, primarily driven by influenza and SARS-CoV-2, followed by RSV. Age and comorbidities significantly impact disease severity, emphasizing the need for targeted prevention and management strategies for RSV in this vulnerable population.

## 1. Introduction

The year 2020 witnessed an unprecedented global health crisis due to the COVID-19 pandemic. In Spain, there has been a 68.5% increase in deaths related to respiratory diseases compared to 2019 [1]. This increase not only led to an overall decline in life expectancy, but also raised concerns about the possible long-term persistence of this trend [2,3,4]. However, the impact of respiratory infections extends beyond the context of the COVID-19 pandemic. Reports such as that of Heppe-Montero et al. [5] suggest that respiratory infections, including those caused by respiratory syncytial virus (RSV), among others, play an important role in morbidity and mortality, especially among elderly patients. As the world′s population ages, the morbidity and mortality associated with respiratory infections continues to increase [6]. 

For many years, the contribution of different viral agents to acute respiratory illnesses in elderly patients was primarily limited to studies on influenza virus. The scarcity of studies was largely due to diagnostic challenges and symptom overlap among different respiratory viruses [7]. However, modern laboratory tools now allow for the identification of respiratory infectious viruses (RIV). Since 2016, there has been increasing interest in RSV, evidenced by efforts by the World Health Organization (WHO) to strengthen global RSV surveillance systems. Nevertheless, much of the available data on RSV disease in adults over 60 years old comes from a limited number of geographic locations, mainly from the United States. Different cultures, climates, and multigenerational household dynamics may significantly influence infection rates [8,9]. There is an urgent need for broader data on infections caused by this virus in the population. In Spain, after the COVID-19 pandemic, the acute respiratory infection surveillance system (SiVIRA) emerged against influenza virus, SARS-CoV-2, and RSV, which demonstrates the growing interest in studying the different etiologies of acute respiratory infections in adults on which we can intervene by changing their evolutionary course, through treatment with antivirals or monoclonal antibodies, and prevention with vaccines [10].

RSV is a typical human pathogen that affects specific population groups as much as or more severely than seasonal influenza [5,11,12]. It is estimated to cause 3.4 million severe cases requiring hospitalization annually [3]. In recent years, RSV has emerged as a leading cause of acute respiratory illness, posing a high risk of severe complications in elderly individuals or those with chronic diseases [5,9,13,14,15]. The gradual decline in innate and adaptive immune system effectiveness associated with aging (immunosenescence) likely contributes to severe RSV illness in the elderly [16,17]. This risk is compounded by high RSV incidence rates observed in specific age groups, such as those over 79 years (40.2 cases/100,000 population) and the 65–79 age group (1.5 cases/100,000 population) [18]. Additionally, RSV does not induce long-lasting immunity, potentially causing multiple reinfections throughout life, which can be life-threatening in patients with underlying conditions [10,19,20,21]. RSV imposes a significant burden on healthcare systems. In Spain, RSV-related hospitalizations cost an estimated €50 million annually to the National Health System [22,23,24]. Therefore, vaccines can play a crucial role, especially in elderly patients or those with comorbidities [17].

Efforts over the past 15 years have been focused on developing various RSV immunoprophylaxis options, such as long-acting monoclonal antibodies for passive immunization of newborns and infants, and vaccines for pregnant women, children, and adults over 60 years old [10]. On 3 May 2023, the US Food and Drug Administration (FDA) approved GSK’s Arexvy vaccine for individuals over 60 years old, which is administered in a single dose. A clinical trial involving nearly 25,000 adults over 60 demonstrated 83.0% efficacy in preventing lower respiratory tract illnesses and 94% efficacy against severe illness, regardless of RSV subtype and underlying comorbidities [16]. On 21 May 2023, Pfizer’s Abrysvo vaccine received FDA approval for individuals over 60 years old and on 21 August 2023, it was approved by the FDA for protection of newborns and young infants through active immunization of pregnant women. Currently, neither vaccine is funded by the Spanish National Health System. Furthermore, new RSV vaccines are being developed by Moderna and Bavarian Nordic, and there is promising potential in using monoclonal antibodies as a therapeutic strategy [25].

Diagnosing and preventing RSV in adult patients, especially the vulnerable, are crucial for reducing morbidity, mortality, and excessive healthcare resource use [12,26]. The aim of this study is to assess the impact of RSV infections in adults over 60 years of age in different communities within the national territory, identifying factors that may be related to greater severity of the disease and developing predictive models that can improve management and the allocation of healthcare resources.

## 2. Materials and Methods

### 2.1. Study Design

We conducted a multicenter retrospective observational study using secondary anonymized data extracted from the electronic health records (EHR) of adult patients aged 60 years and older admitted between 1 October 2023 and 31 March 2024 at 18 university hospitals of the HM group. HM Hospitales (HM) is a network of 21 private university hospitals with a presence across Spain in the regions of Madrid (Madrid, Madrid Río, Montepríncipe, Nuevo Belén, Puerta del Sur, Rivas, Sanchinarro, Torrelodones, and Vallés), León (Regla and San Francisco); Cataluña (Nens, Nou Delfos, and Sant Jordi), Galicia (Belen, Esperanza, Modelo, and Rosaleda) and Andalucía (El Pilar, Gálvez, Málaga, and Santa Elena). All patients had a diagnosis of respiratory infection due to RIV confirmed by PCR or antigen test.

### 2.2. Ethical Approval and Informed Consent

The study was conducted in accordance with the Declaration of Helsinki and approved by the Ethics Committee of HM Hospitales (protocol code 24.04.2344-GHM and date of approval: 8 May 2024) for studies involving humans. Moreover, this research is a retrospective study using anonymized data, which involves no direct patient intervention; therefore, individual consent to participate is not necessary.

### 2.3. Variables Collected

We collected sociodemographic variables such as admission and discharge dates from both regular wards and intensive care units (ICU), date of birth, gender, and mortality status. Additionally, we retrieved International Classification of Diseases, 10th Revision (ICD-10) codes to extract various variables, including smoking status, alcohol consumption, associated pathologies (such as asthma, diabetes, obesity, hypertension, chronic obstructive pulmonary disease (COPD), renal failure, cardiac diseases, neoplasms), use of medical procedures (high-flow nasal cannula, oxygen therapy, invasive ventilation), and symptoms present during the episode (pneumonia, upper respiratory tract infection (URTI), lower respiratory tract infection (LRTI), acute respiratory distress syndrome). The Charlson Comorbidity Index was calculated for each patient during each admission using the Comorbidity Package [27]. We also collected variables on pharmacological treatments administered during hospitalization (antibiotics, anticoagulants, antihypertensives, antivirals, diuretics). Data were extracted and anonymized from local EHRs to protect patient confidentiality and privacy, then combined into a unified harmonized dataset.

### 2.4. Statistical Analyses

First, a descriptive and comparative analysis of sociodemographic variables was conducted. Continuous variables were presented as mean and standard deviation (SD). Normality of continuous variables was assessed using the Shapiro–Wilk test or Kolmogorov–Smirnov test, while homoscedasticity was tested using Levene’s test. Parametric tests (t-test or ANOVA) or non-parametric tests (Kruskal–Wallis or Wilcoxon rank-sum test) were applied based on adherence to these assumptions. Categorical variables were compared using the chi-square test or Fisher’s exact test, as appropriate. Total admission cases were used to calculate incidence rate ratios and corresponding 95% confidence intervals (CI). Finally, to identify potential predictors of severity in RVI-related hospital admissions, a multi-state model was constructed. Variables were selected using univariate modelling for each predictive variable available for each state (ICU admission, mortality). In addition, we assessed the proportional hazards criterion and whether in the case of continuous variables they followed a non-linear or time-dependent distribution. A significance level of *p* < 0.05 was used for all analyses. For multiple comparisons, corrections (Bonferroni method) were utilized to adjust for inflated Type I error rates. Data manipulation was performed by the function contained in the tidyverse package [28]. Rate analyses were performed using the epiR package [29] and proportional hazards and multistate cox regression testing were performed using the survival and mstate packages [30,31], respectively. Data processing and statistical analyses were performed using *R* [32], version 4.3.1.

## 3. Results

### 3.1. Respiratory Disease Hospitalizations in Elder Patients

From 1 October 2023, to 31 March 2024, a total of 22,201 episodes involved hospitalizations of patients aged 60 and older, across HM Hospitales centers. Among these, 3258 episodes (14.7%) were selected where patients were diagnosed with respiratory pathology, including URTI, LRTI, flu, or pneumonia. All patients underwent viral detection testing (positive antigen test or PCR for viruses) during their hospital stay. Out of the total respiratory pathology diagnoses, 1933 cases tested positive for at least one RVIs, with 31.2% of these patients positive for SARS-CoV-2, 19.4% for influenza virus, 7.0% for RSV, 6.0% for rhinovirus, 1.7% for metapneumovirus, and the remaining 4.9% positive for other RVI such as adenovirus, bocavirus, enterovirus, and parainfluenza (Table 1). Among the total admissions (1933 in total), 1704 (88.2%) were attributed to a single RVIs, 152 (7.9%) to dual RVI co-infections, and 77 (4.0%) to infections involving more than two RVIs. The impact of each RVI of interest relative to the total number of unique episodes where any RVI was detected was measured. First of all, SARS-CoV-2 was most prevalent (1016; 52.6%), followed by influenza virus (632; 32.7%), and RSV (229; 11.8%).

The admission rates, ICU admission rates, and mortality rates with presence of RIV were calculated per 10,000 admissions of patients aged 60 and above for the season spanning from 1 October 2023, to 31 March 2024 (Figure 1; Table A1). It was observed that SARS-CoV-2 admissions were highest in all months of the analysed season, except for January 2024, where it was surpassed in all three analyses (admission, ICU admission, and mortality rates). In second place, influenza virus was consistently the second most prevalent virus throughout the season. RSV ranked third for most of the season, peaking during the months of November to January.

### 3.2. Hospitalizations by RVI for Each Age Group and Outcome

To assess the impact of predominant RVI (SARS-CoV-2, influenza virus and RSV) on hospital admissions across age groups, admission rates per 10,000 admissions along with 95% confidence intervals (CI), ICU admissions, and mortality rates were analysed (Figure 2; Table A2). The admission rate increased as the age of the group increased with any of the viral infections studied (SARS-CoV-2: 333.4 [95% CI: 295.0–375.2] to 651.6 [95% CI: 532.1–788.4]; influenza virus: 169.8 [95% CI: 142.6–200.7] to 518.6 [95% CI: 412.1–643.1] and RSV: 69.2 [95% CI: 52.2–90.0] to 246.0 [95% CI: 173.8–337.5]). 

This trend significantly surpassed the average admission rate for SARS-CoV-2 (518.9), influenza virus (347.1), and RSV (139.7) in age groups 80 years and older. SARS-CoV-2 had the highest admission rate, followed by influenza virus and RSV.

Regarding ICU admission or mortality rates, there were no significant differences observed among the RVIs. However, the average ICU admission rate was higher for influenza virus (127.14) compared to SARS-CoV-2, followed by RSV (86.7 and 41.3, respectively). Specific ICU admission rates were higher for SARS-CoV-2 infections (101.4 [95% CI: 52.5–176.4]), for RSV (166.7 [95% CI: 76.5–314.0]) in patients aged 70–79 years, and for influenza virus infections in patients aged 80–89 years (166.7 [95% CI: 76.5–314.0]). When evaluating mortality rates per 10,000 admissions for patients who died, the ranking by average mortality rate was as follows: influenza virus (333.5), SARS-CoV-2 (320.3), and RSV (173.5). Specific mortality rates for SARS-CoV-2 infections were higher in patients aged 70–79 years (279.7 [95% CI: 76.7–700.7]), for influenza virus infections in patients aged 80–89 years (347.83 [95% CI: 151.3–673.8]), and for RSV infections in patients aged 90 years and older (363.6 [95% CI: 134.6–774.65]).

### 3.3. Sociodemographic and Clinical Characteristics

Out of a total of 1933 unique episodes of patients positive for RVIs, 1721 (89.0%) were single infections. These single infections corresponded to SARS-CoV-2 (936 admissions; 54.39%), influenza virus (500; 29.1%), RSV (132; 7.7%), rhinovirus (77; 4.5%), metapneumovirus (36; 2.1%), and other minority viruses (40; 2.3%). Focusing only on admissions where the main viruses were presented (1568 admissions; 936 of SARS-CoV-2, 500 of influenza virus and 132 of RSV), the sociodemographic characteristics of our sub-sample had a mean admission age of 77.1 ± 9.4 years, with a mean stay of 7.6 ± 8.2 days and admissions ending in ICU admission (39; 2.5%) had a mean stay of 4.7 ± 7.5 days.

When we focus on the characteristics of the admissions of each specific virus (Table 2), females were predominant across all RVI of interest, with a significantly (*p* = 0.030) higher representation in RSV admissions (85 admissions; 64.4%) and lower representation in SARS-CoV-2 (522; 55.8%) and influenza admissions (259; 51.8%). Mean age at admission varied significantly (*p* = 0.017), being significantly lower for SARS-CoV-2 patients (76.6 ± 9.2) and higher for RSV patients (79.0 ± 10.4). Hospital length stay also showed significant differences (*p* < 0.001), with significantly longer stays for RSV (8.6 ± 5.9 days) and influenza virus (8.6 ± 10.4 days) compared to SARS-CoV-2 patients (6.6 ± 6.4 days). In terms of admission outcome, patients with RSV (6; 4.5%) had a significantly higher mortality rate than those with SARS-CoV-2 (13; 1.4%). Active smokers were also significantly more prevalent in RSV (18; 13.6%), and influenza (62; 12.4%) admissions compared to SARS-CoV-2 (64; 6.8%).

Evaluation of comorbidities revealed that the proportion of admissions with obesity were significantly lower (*p* < 0.001) when SARS-CoV-2 is presented (48; 5.1%) in comparison with influenza (64; 12.8%) and RSV (16; 12.1%). COPD prevalence showed significant differences (*p* = 0.008), being higher in RSV patients (40; 30.3%) and lower in SARS-CoV-2 patients (152; 16.2%). Lastly, the Charlson Comorbidity Index showed significant differences (*p* < 0.001), being highest in RSV admissions (3.3 ± 3.1) and lowest in SARS-CoV-2 patients (2.1 ± 2.9).

Symptom presentation varied significantly among different RVI (*p* ≤ 0.007). Pneumonia was more frequent in RSV (20; 15.2%), and less common in SARS-CoV-2 (81; 8.7%) and influenza (34; 6.8%) patients. LRTI and URTI were significantly more frequent (*p* = 0.007 and *p* < 0.001, respectively) for RSV admissions. 

Regarding treatments, significant differences were observed in the use of antibiotics and antivirals among the admissions where the main viruses are presented (*p* < 0.001). Antibiotic use was higher in RSV (83; 62.9%), and lower in SARS-CoV-2 patients (257; 27.5%). Antiviral use was significantly higher in influenza (162; 32.4%), and lower in SARS-CoV-2 (18; 1.9%) and RSV (8; 6.1%) cases.

### 3.4. Assessing the Effect of Factors on Episode Severity

We assessed the impact of various severity-related factors (ICU, exitus) using a multi-state model. Our dataset defined three states: Admitted, ICU, and exitus. All patients started in the Admitted state, with some transitioning to ICU and others directly to exitus. Additionally, transitions from ICU to exitus are possible (Figure 3). 

Prior to constructing the multi-state Cox model, we checked whether each univariate predictor variable available was significant (*p* < 0.05), as well as assessing whether they met the proportional hazards criterion. In addition, it was assessed whether the continuous variables followed a non-linear distribution or were time dependent. In neither case were these assumptions made (*p* > 0.05). After selecting the most relevant variables, we proceeded with the multi-state Cox model analysis (Table 3).

Among all evaluated variables, age at admission showed significant differences across all transitions. For the transition from Admission to ICU, the hazard ratio (HR) was 0.9 (95% CI: 0.9–0.9, *p* < 0.001), indicating that older age decreases the likelihood of transfer to ICU. Conversely, for the transition from Admission to exitus, the HR was 1.2 (95% CI: 1.1–1.3, *p* < 0.001), suggesting that older age increases the likelihood of mortality directly from the admission state. Similarly, for the transition from ICU to exitus, the HR was 1.1 (95% CI: 1.0–1.2, *p* < 0.001), indicating a significantly higher risk of death in ICU with increasing age.

Furthermore, the presence of RSV proved to be a significant risk factor in the transition from admission to ICU with an HR of 2.2 (95% CI: 1.1–4.3, *p* = 0.037), indicating that RSV presence increases the likelihood of ICU admission by 2.2 times compared to those without RSV. Meanwhile, the presence of antibiotics was a protective factor, with an HR of 0.5 (95% CI: 0.2–0.9, *p* = 0.041). In the transition from admission to exitus, the presence of a higher score increased by 40% the likelihood of exitus from admission, with an HR of 1.4 (95% CI: 1.2–1.6, *p* < 0.001). Anticoagulants and diuretic treatment were significant protective factors in the transition from admission to exitus, with HRs of 0.1 (95% CI: 0.0–0.3, *p* = 0.001) and 0.3 (95% CI: 0.1–0.9, *p* = 0.038), respectively.

## 4. Discussion

This study offers a comprehensive analysis of RVI in adults aged 60 and older, emphasizing their significant impact on morbidity and mortality in this demographic. Our research identifies SARS-CoV-2, influenza, and RSV as the primary viral agents leading to hospitalizations due to respiratory conditions in this cohort. This analysis sheds light on the evolving landscape of viral infections and their implications for public health strategies targeting older adults.

The high incidence of SARS-CoV-2 observed in our sample can be attributed to the analysis period, which followed the COVID-19 epidemics, during which virus testing was paramount. Widespread testing facilitated the early detection and isolation of infected individuals, thereby influencing hospitalization rates. However, this study goes beyond the immediate effects of SARS-CoV-2 to examine the broader spectrum of viral infections affecting older adults.

Our findings reveal a substantial impact of RSV, accounting for 11.8% of respiratory infections in admissions for adults aged 60 and older. This incidence rate of RSV in our study is notably higher than the national average of 7.9% reported in 2024 [33], making it the third most common viral cause after SARS-CoV-2 and influenza. The heightened incidence of RSV underscores the need for increased awareness and targeted interventions to address this often-overlooked virus.

RSV, traditionally considered less impactful in older adults, is associated with higher complication rates during hospitalization, such as pneumonia and LRTIs. These complications frequently require additional treatments, including antibiotics and diuretics. Notably, patients with RSV experienced longer hospital stays and higher mortality rates compared to those with SARS-CoV-2 and influenza. Furthermore, RSV infection significantly increased the risk of ICU admission, with a 2.2-fold higher likelihood of transfer to intensive care.

The impact of RSV is particularly pronounced in older adults with specific risk factors, such as smoking and comorbidities like COPD and heart failure. Advanced age and underlying health conditions significantly elevate the risk of severe disease, highlighting the need for tailored healthcare strategies for these vulnerable groups [5,13].

Respiratory viral infections impose a substantial socioeconomic burden due to increased healthcare utilization and associated costs. While scientific focus is primarily on influenza and SARS-CoV-2, other respiratory viruses like RSV can be equally or more detrimental, particularly in older adults [5,9,13,14,15]. The circulation and aggressiveness of these viruses have shifted post-COVID-19, with notable changes in their seasonality [21,34]. This necessitates robust epidemiological research and adaptive immunization strategies to protect the most vulnerable populations effectively.

One of the strengths of this study is its focus on the effects of RSV in adults aged 60 and older in Spain, a relatively underexplored area. This provides significant value as we demonstrate that older adults, particularly those with specific comorbidities, are susceptible and sometimes more vulnerable to RSV than to other more widely recognized viruses. However, the study also has limitations. The use of varied detection techniques for respiratory viruses, such as PCR and antigen tests, might affect the heterogeneity of detection sensitivity and specificity [35]. Additionally, these detection tests primarily target the major viruses evaluated in this study and may overlook co-infections with less common viruses. There is a possibility that a higher percentage of co-infections were diagnosed in more severely ill patients, who underwent more extensive testing due to their worse prognosis.

Further research is essential to confirm whether there is a higher morbidity and mortality associated with these viruses. This study reinforces the need to investigate the etiology of respiratory infections in older adults to identify factors associated with disease severity. Early diagnosis and treatment, coupled with appropriate preventive strategies, could significantly improve healthcare quality, reduce morbidity and mortality, and decrease healthcare resource utilization. Effective public health policies and continuous surveillance are vital for protecting older adults from the evolving threat of respiratory viral infections [12,26]. Given the availability of vaccines to prevent severe COVID-19, influenza, and RSV from 2024, public health decisions will be crucial. Future strategies will be instrumental in preventing severe RSV infections in this population. It is essential to emphasize the importance of vaccination, especially since these vaccines are now available in Spain.

## 5. Conclusions

In conclusion, this study highlights the heightened vulnerability of older adults, particularly those with comorbidities, to RSV, underscoring the need for increased awareness, early and accurate diagnosis, and tailored healthcare interventions. The findings also emphasize the critical role of vaccination in alleviating the burden of RVI in this population, reinforcing the importance of preventive strategies to reduce hospitalizations, severe complications, and mortality among older adults.

## Figures and Tables

**Figure 1 geriatrics-09-00145-f001:**
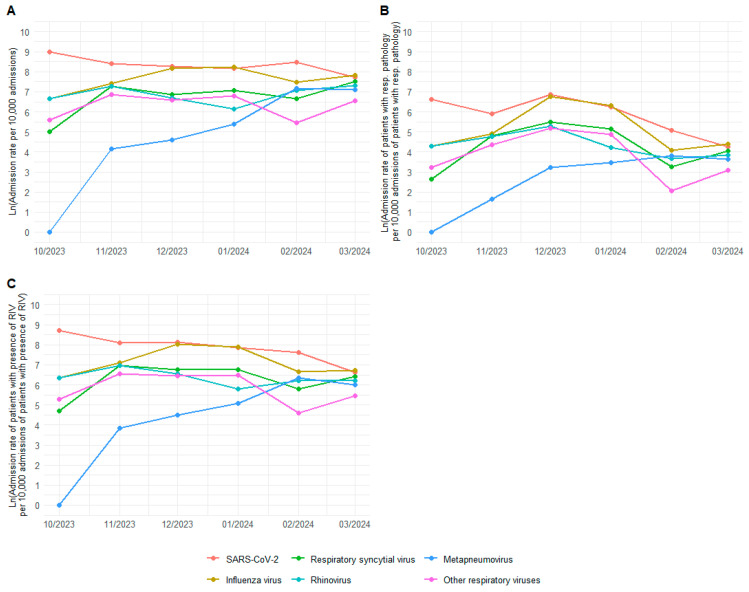
Transformed admission rates for each of the viruses for all total admissions with a minimum age of 60 years (**A**), admissions with respiratory pathology (**B**), and admissions with a positive sample for any of the viruses of interest (**C**) per 10,000 admissions in the season from 1 October 2023 to 31 March 2024.

**Figure 2 geriatrics-09-00145-f002:**
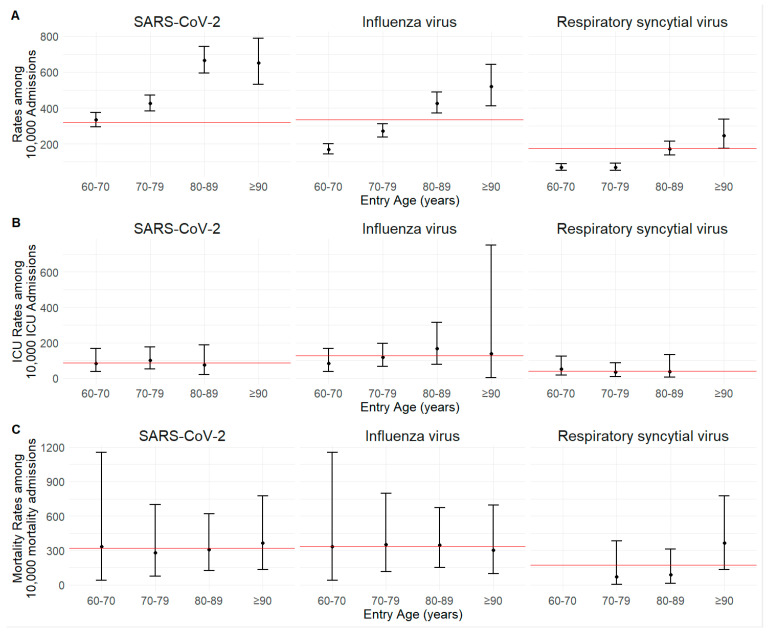
Admission (**A**), ICU admission (**B**), and mortality (**C**) rates, and 95% confidence intervals, of patients aged 60 years and older for RVI during the season from 1 October 2023 to 31 March 2024. Rates are expressed per 10,000 admissions of patients with these characteristics and are broken down by age group. The red line symbolises the average rate per 10,000 patient admissions for each virus.

**Figure 3 geriatrics-09-00145-f003:**
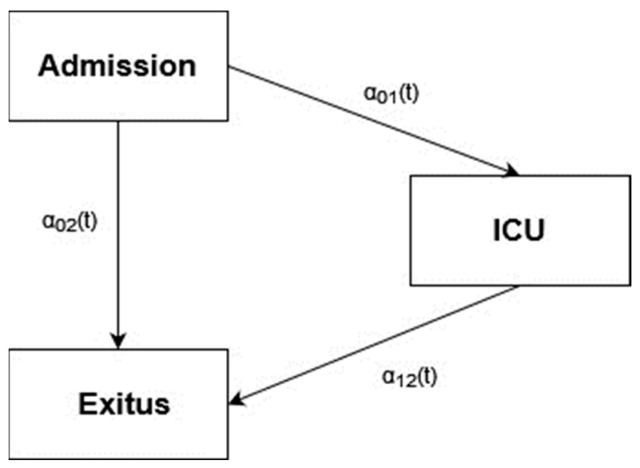
Multi-state model Admission-ICU-exitus.

**Table 1 geriatrics-09-00145-t001:** Impact of RVI on total admissions for respiratory pathology and positive admissions with RVI detection.

Items	Number of Admissions (Due to Respiratory Pathology ^2^ (n = 3258), for Any of the RVI of Interest (n = 1933)) ^3^
Including co-infections	
SARS-CoV-2	1016 (31.2%, 52.6%)
Influenza	632 (19.4%, 32.7%)
RSV	229 (7.0%, 11.8%)
Rhinovirus	194 (6.0%, 10.0%)
Other viruses ^1^	160 (4.9%, 8.3%)
Metapneumovirus	54 (1.7%, 2.8%)
Single infections	
SARS-CoV-2	936 (28.7%, 48.4%)
Influenza	500 (15.4%, 25.9%)
RSV	132 (4.1%, 6.8%)
Rhinovirus	77 (2.4%, 4.0%)
Other viruses ^1^	40 (1.2%, 2.1%)
Metapneumovirus	36 (1.1%, 1.9%)
Major RSV co-infections	
RSV + rhinovirus + other viruses	42 (1.3%, 2.2%)
Influenza + RSV	36 (1.1%, 1.9%)
Influenza + RSV + rhinovirus + other viruses	36 (1.1%, 1.9%)
Influenza + rhinovirus	31 (1.0%, 1.6%)
SARS-CoV-2 + RSV+ rhinovirus + other viruses	24 (0.7%, 1.2%)

^1^ The classification “other viruses” includes positive episodes for adenovirus, bocavirus, enterovirus, and parainfluenza. ^2^ Includes only episodes with ICD-10 referring to respiratory pathology (Infectious diseases of the upper (J00-J06) and lower (J20-J22) respiratory tract, flu and pneumonia (J10-J18), or having a positive test for one of the RVI of interest. ^3^ Some patients may have had co-infection, so the sum does not add up to the total.

**Table 2 geriatrics-09-00145-t002:** Descriptive analysis of admissions with RIV for each with single infections.

Items	SARS-CoV-2 (n1 = 936)	Influenza (n2 = 500)	RSV (n3 = 132)	*p*-Value
Sex [Female]	522 (55.8%) ↓	259 (51.8%) ↓	85 (64.4%) ↑	0.030
Age at admission	76.6 (±9.2) ↓	77.7 (±9.5)	79.0 (±10.4) ↑	0.006
Length of stay	6.6 (±6.4) ↓	8.6 (±10.4) ↑	8.61 (±5.9) ↑	<0.001
ICU	17 (1.8%)	19 (3.8%)	3 (2.3%)	0.070
ICU days	5.2 (±7.1)	4.7 (±8.5)	1.3 (±0.6)	0.435
Exitus	13 (1.4%) ↓	11 (2.2%)	6 (4.5%) ↑	0.040
Smoker	64 (6.8%) ↓	62 (12.4%) ↑	18 (13.6%) ↑	0.008
Former smoker	193 (20.6%)	114 (22.8%)	34 (25.8%)	0.995
Alcoholism	28 (3.0%)	26 (5.2%)	3 (2.3%)	0.098
Comorbidities				
Asthma	41 (4.4%)	32 (6.4%)	7 (5.3%)	0.428
Diabetes	157 (16.8%)	92 (18.4%)	22 (16.7%)	0.468
Obesity	48 (5.1%) ↓	64 (12.8%) ↑	16 (12.1%) ↑	<0.001
Hypertension	443 (47.3%)	266 (53.2%)	84 (63.6%)	0.715
COPD	152 (16.2%) ↓	117 (23.4%)	40 (30.3%) ↑	0.008
Kidney failure	128 (13.7%)	84 (16.8%)	27 (20.5%)	0.566
Heart failure	154 (16.5%)	95 (19.0%)	40 (30.3%)	0.053
Neoplasia	188 (20.1%)	99 (19.8%)	35 (26.5%)	0.490
Charlson Index	2.13 (±2.9) ↓	2.74 (±3.3)	3.33 (±3.1) ↑	<0.001
Use of health services				
High-flow nasal goggles	3 (0.3%)	0 (0.0%)	0 (0.0%)	0.199
Oxygen therapy	17 (1.8%)	21 (4.2%)	7 (5.3%)	0.340
VMI	5 (0.5%)	3 (0.6%)	1 (0.8%)	0.908
Symptoms				
Pneumonia	81 (8.7%) ↓	34 (6.8%) ↓	20 (15.2%) ↑	<0.001
LRTI	6 (0.6%) ↓	16 (3.2%)	8 (6.1%) ↑	0.007
URTI	4 (0.4%) ↓	1 (0.2%) ↓	18 (13.6%) ↑	<0.001
Respiratory distress syndrome	7 (0.7%)	1 (0.2%)	0 (0.0%)	0.079
Treatments				
Antibiotics	257 (27.5%) ↓	240 (48.0%)	83 (62.9%) ↑	0.001
Anticoagulants	236 (25.2%)	202 (40.4%)	66 (50.0%)	0.961
Antihypertensives	14 (1.5%)	12 (2.4%)	0 (0.0%)	0.132
Antivirals	18 (1.9%) ↓	162 (32.4%) ↑	8 (6.1%) ↓	<0.001
Diuretics	119 (12.7%)	115 (23.0%)	45 (34.1%)	0.057

Data are shown as n (%), mean (± standard deviation). NA: Not available. LRTI: Lower respiratory tract infection, URTI: Upper respiratory tract infection. The direction of the pairwise post hoc significant results (*p* < 0.05) is indicated (↑, ↓).

**Table 3 geriatrics-09-00145-t003:** Cox multistate inpatient-ICU-exitus model.

Item	Admission-ICU Transition		Admission-Exitus Transition		ICU-Exitus Transition	
	Hazard Ratios (95% CI)	*p*-Value	Hazard Ratios (95% CI)	*p*-Value	Hazard Ratios (95% CI)	*p*-Value
Age entry	0.9 (0.9–0.9)	<0.001	1.2 (1.1–1.3)	<0.001	1.1 (1.0–1.2)	<0.001
Neoplasia	0.9 (0.3–2.6)	0.854	0.5 (0.2–1.4)	0.203	0.5 (0.1–3.0)	0.436
Kidney insufficiency	0.8 (0.4–1.6)	0.596	0.9 (0.4–2.1)	0.860	1.9 (0.4–8.6)	0.406
Heart failure	1.9 (0.8–4.5)	0.159	0.5 (0.21–1.4)	0.200	0.8 (0.1–4.5)	0.797
Charlson score	0.9 (0.8–1.1)	0.274	1.4 (1.2–1.6)	<0.001	1.6 (0.9–2.8)	0.081
Pneumonia	0.7 (0.4–1.4)	0.311	1.2 (0.5–3.2)	0.689	0.4 (0.1–2.9)	0.364
Antibiotics	0.5 (0.2–0.9)	0.041	0.6 (0.0–9.0)	0.710	3.7 (0.5–25.5)	0.186
Anticoagulants	1.9 (0.5–7.1)	0.315	0.1 (0.0–0.3)	0.001	0.3 (0.0–2.7)	0.308
Diuretics	0.8 (0.3–1.6)	0.472	0.2 (0.1–0.9)	0.038	0.2 (0.0–2.0)	0.151
Influenza	1.1 (0.6–2.1)	0.723	1.6 (0.6–4.3)	0.365	0.8 (0.2–3.8)	0.783
RSV	2.2 (1.1–4.6)	0.037	0.9 (0.3–3.3)	0.895	NA (NA–NA)	NA

CI: confidence intervals. NA: not available.

## Data Availability

The data supporting the reported results cannot be shared due to ethical restrictions.

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
