# Peer review of "Impact of Respiratory Syncytial Virus (RSV) in Adults 60 Years and Older in Spain"

_geriatrics, 2024, doi:10.3390/geriatrics9060145_

Round 1

Reviewer 1 Report

Comments and Suggestions for Authors

Very good information overall, I really appreciate the way the methods are presented.

However, I have two main questions: the first is why this title when in the results the greatest impacts are found for Sars-CoV-2 and Influenza virus? The second is whether or not the RSV vaccine is used... If so, it would be good to mention it.

A few other comments: 

- Lines 26 and 27: Can you put the values that show this?

- Line 111: Delete the dot (.) after “hospitals”.

- Table 1: It would be good to choose between Influenza virus and influenza, Covid and Sars-CoV-2

- Line 243: What is VRS? Choosing between VRS and RSV

Author Response

Reviewer 1

Comments and Suggestions for Authors

Very good information overall, I really appreciate the way the methods are presented. However, I have two main questions: the first is why this title when in the results the greatest impacts are found for Sars-CoV-2 and Influenza virus?

The focus of this study on Respiratory Syncytial Virus (RSV), despite the fact that the highest hospitalization rates were found for SARS-CoV-2 and influenza, is justified by the relative scarcity of studies addressing the impact of RSV in adults 60 years and older. RSV is often overshadowed by other respiratory viruses like SARS-CoV-2 and influenza, but its effects on older adults are increasingly recognized as a significant public health concern. Although less prevalent in the data, RSV poses a substantial risk for severe complications, such as increased ICU admissions, especially among elderly populations with pre-existing conditions. Furthermore, unlike SARS-CoV-2 and influenza, RSV does not confer long-lasting immunity, making recurrent infections more likely and dangerous in older adults.

Given the recent advancements in RSV vaccination and the growing interest in its epidemiology, this study aims to fill the gap in the literature by focusing on RSV, which, while less prevalent, is a leading cause of hospitalization and severe outcomes in older adults. By concentrating on RSV, the study provides crucial insights for future preventive measures, including the deployment of vaccines and treatments specifically targeting this virus in a vulnerable population. Thus, the title highlights the central objective of improving understanding of RSV in this demographic, where existing data remain limited.

The second is whether or not the RSV vaccine is used... If so, it would be good to mention it.

In response to the second point, at the time of the study, the RSV vaccine had not yet been administered to this population in Spain. For this reason, it was not mentioned in the manuscript.

A few other comments: 

- Lines 26 and 27: Can you put the values that show this?

Added: Hospitalization rates increased with age for SARS-CoV-2 (333.4 [95% CI: 295.0 – 375.2] to 651.6 [95% CI: 532.1 – 788.4]), influenza (169.8 [95% CI: 142.6 – 200.7] to 518.6 [95% CI: 412.1 – 643.1]), and RSV (69.2 [95% CI: 52.2 – 90.0] to 246.0 [95% CI: 173.8 – 337.5]), with SARS-CoV-2 showing the highest rate, followed by influenza and RSV.

- Line 111: Delete the dot (.) after “hospitals”.

Deleted

- Table 1: It would be good to choose between Influenza virus and influenza, Covid and Sars-CoV-2

Homogenised

- Line 243: What is VRS? Choosing between VRS and RSV

Homogenised to RSV

Reviewer 2 Report

Comments and Suggestions for Authors

Jimeno et al. described the rate of RVIs and their impact on the admission and the admission outcome over a six month period from Spain. Despite the short period of the study, the multi-centre design and the large number of samples give the study its value.

I have the following minor comments:

1. The title is not reflecting the contents of the paper. it should mention the three viruses (SARS-CoV-2, Influenza, and RSV)

2. There is discrepancy in the first part of the results in the abstract and the result section in the text. please revise it accurately regarding the number and percentages of admissions and RVIs.

3. provide information about HM hospitals in the methods 

4. the abbreviations RVI and RSV are misspelled sometimes in the manuscript.

5. line 116 add test after antigen.

6. line 167-169. the sentence is not understandable

7. table 1. the line number is overlapping with the text in the table

8. Abbreviations such as ICD-10 and HR should be mentioned as full first

9. figure 2. the calculation is made out of 10,000 admissions. but in 2A and 2B, it is mentioned 1000

Comments on the Quality of English Language

In general, the Manuscript is readable and understandable, however, very few sentences needs attention to grammar and word repetition.  

I believe, a thorough read through by the authors will solve the problem

Author Response

Comments and Suggestions for Authors

Jimeno et al. described the rate of RVIs and their impact on the admission and the admission outcome over a six month period from Spain. Despite the short period of the study, the multi-centre design and the large number of samples give the study its value.

I have the following minor comments:

  1. The title is not reflecting the contents of the paper. it should mention the three viruses (SARS-CoV-2, Influenza, and RSV)

The title focuses specifically on RSV to underscore its often-overlooked significance in older adults, despite the presence of SARS-CoV-2 and influenza. While these viruses have higher hospitalization rates, our study aims to illuminate the substantial impact of RSV on this demographic, which has been under-researched. By emphasizing RSV, the title reflects our objective of addressing the knowledge gap regarding its effects and complications in older adults, particularly in light of recent advancements in RSV vaccination. This targeted approach is essential for developing effective public health strategies and interventions tailored to this vulnerable population. Including the other viruses in the title would detract from this focused investigation and our goal of enhancing understanding of RSV's unique challenges in older adults.

  1. There is discrepancy in the first part of the results in the abstract and the result section in the text. please revise it accurately regarding the number and percentages of admissions and RVIs.
  2. provide information about HM hospitals in the methods 

Specified: The HM Hospitals (HM) is a network of 21 private University Hospitals with presence across Spain in the regions of Madrid (Madrid, Madrid Río, Montepríncipe, Nuevo Belén, Puerta del Sur, Rivas, Sanchinarro, Torrelodones and Vallés),  León (Regla and San Francisco); Cataluña (Nens, Nou Delfos and Sant Jordi), Galicia (Belen, Esperanza, Modelo and Rosaleda) and Andalucía (El Pilar, Gálvez, Málaga, Santa Elena).

  1. the abbreviations RVI and RSV are misspelled sometimes in the manuscript.

They have been corrected

  1. line 116 add test after antigen.

Added

  1. line 167-169. the sentence is not understandable

The sentence was chaged to: “The impact of each RVIs was analysed by a person-time approach”. Now this sentence is simple and understandable.

  1. table 1. the line number is overlapping with the text in the table.

We added a space between the text and the table 1

  1. Abbreviations such as ICD-10 and HR should be mentioned as full first

Defined ICD-10in line 124 and HR in line 312

  1. figure 2. the calculation is made out of 10,000 admissions. but in 2A and 2B, it is mentioned 1000

Modified the figure 2 typo.

Comments on the Quality of English Language

In general, the Manuscript is readable and understandable, however, very few sentences needs attention to grammar and word repetition.  

I believe, a thorough read through by the authors will solve the problem

Thank you for your feedback on the quality of the English language in our manuscript. We have taken your comments to heart and have thoroughly reviewed the text, correcting all identified grammatical issues and reducing word repetition. We appreciate your insights, which have helped us enhance the clarity and quality of our work.

Reviewer 3 Report

Comments and Suggestions for Authors

Please check the sentence 'RSV is a typically human pathogen, impacting specific population groups comparable to or more than seasonal influenza (5, 11, 12).' It is unclear what 'impacting more than influenza' would mean.

Line 85: Since the uptake of the RSV vaccines is not widespread yet, I would add the word 'can' in the sentence: Therefore, vaccines can play a crucial role, especially in elderly patients or those with comorbidities (17).

Lines 94-96, please change the order of the two sentences: 'On May 18, 2023, Pfizer's Abrysvo vaccine received FDA approval for individuals over 60 years old and on August 21, 2023 it was approved by the FDA for protection of newborns and young infants through active immunization of pregnant women.'

Line 162 and other places. Please, check the use of the singular 'RVI' or plural 'RVIs' throughout the text. In this sentence it should be 'at least one RVI'.

Table 1 – The number of cases are repeated every time in the two columns. I recommend to indicate the numbers as 'N (% of total admissions, % of confirmed RVI). For example: 'SARS-CoV- 2            1,016 (31.2%, 51.6%)'

Figure 1, please change 'income' to 'admissions' on the y-axes. In addition, correct 'income' to 'admission' or 'admissions' throughout the text. What is 'VR' on the y-axis in panel C? If 'VR' is 'RIV' then please change to 'RIV'. Is the log scale in the y-axes a 2log or a 10 log. Please, correct to 2Log in the figure if needed.

Lines 220-222. The sentences seem to contradict each other. The first sentence mentions no differences, while the next sentence points out a difference for ICU admissions. Please, amend the sentences.

Line 226: Please correct typo: '10,000' instead of '100,000'.  

Please, check whole document and table 2 for 'VRS' and correct to 'RSV'

Line 25): Small typo: 'SARS-CoV-2' instead of 'SARS-CoV2'

Author Response

Reviewer 3

Comments and Suggestions for Authors

Please check the sentence 'RSV is a typically human pathogen, impacting specific population groups comparable to or more than seasonal influenza (5, 11, 12).' It is unclear what 'impacting more than influenza' would mean.

We have modified that sentence by “RSV is a typically human pathogen that affects specific population groups as much as or more severely than seasonal influenza”

Line 85: Since the uptake of the RSV vaccines is not widespread yet, I would add the word 'can' in the sentence: Therefore, vaccines can play a crucial role, especially in elderly patients or those with comorbidities (17).

Added

Lines 94-96, please change the order of the two sentences: 'On May 18, 2023, Pfizer's Abrysvo vaccine received FDA approval for individuals over 60 years old and on August 21, 2023 it was approved by the FDA for protection of newborns and young infants through active immunization of pregnant women.'

Changed

Line 162 and other places. Please, check the use of the singular 'RVI' or plural 'RVIs' throughout the text. In this sentence it should be 'at least one RVI'.

Change to RVI

Table 1 – The number of cases are repeated every time in the two columns. I recommend to indicate the numbers as 'N (% of total admissions, % of confirmed RVI). For example: 'SARS-CoV- 2               1,016 (31.2%, 51.6%)'

Changed, removing the redundancy

Figure 1, please change 'income' to 'admissions' on the y-axes.

Modified

In addition, correct 'income' to 'admission' or 'admissions' throughout the text.

Modified

What is 'VR' on the y-axis in panel C? If 'VR' is 'RIV' then please change to 'RIV'. Is the log scale in the y-axes a 2log or a 10 log. Please, correct to 2Log in the figure if needed.

Thank you for your feedback. We have clarified that 'VR' refers to 'RIV' and have updated the label accordingly. The logarithmic scale on the y-axis uses the default in R, which is the natural logarithm (base e). The use of the natural logarithm (ln) is justified in this context as it allows for a clearer representation of the exponential growth of admissions, enabling easier comparison of growth rates and trends over time. So we have changed the y axis to ln.

Lines 220-222. The sentences seem to contradict each other. The first sentence mentions no differences, while the next sentence points out a difference for ICU admissions. Please, amend the sentences.

Line 226: Please correct typo: '10,000' instead of '100,000'. 

Changed

Please, check whole document and table 2 for 'VRS' and correct to 'RSV'

Modified

Line 25): Small typo: 'SARS-CoV-2' instead of 'SARS-CoV2'

Modified